# Corrosion Hazards in Urban Infrastructure Structures Using the Example of the Al Bayt Stadium in Katar

Agnieszka Krolikowska [1] and Pier Luigi Bonora [2,*]

1   Road and Bridge Research Institute, 03-302 Warsaw, Poland; akrolikowska@ibdim.edu.pl
2   AITIVA—Associazione Tecnici Industria Vernici Affini, 29122 Piacenza, Italy
*   Correspondence: bonorapierluigi@gmail.com; Tel.: +39-34835-56332

**Abstract:** A significant aspect of corrosion failures and catastrophes originate from trivial mistakes in either the choice or connection of different materials, as well as from inaccurate evaluation of the compatibility between materials and the environment. The example shown in the present paper summarizes several wrong solutions due to a lack of knowledge of the basic rules of corrosion control. By chance, the consequences of these errors already appeared during construction; therefore, they were not able to cause damage during operation. This paper is the third in a series devoted to enhancing the need for professional corrosion control design for infrastructures.

**Keywords:** corrosion failure; mistakes; stadium

## 1. Introduction

The late XX century is characterized by the massive exploitation of what was handed down from the industrial and social revolutions of the XIX century, with a few exceptions: the semiconductor, atomic energy, antibiotics, and polymers are among the main examples.

Because of the need to handle an enormous number of materials, it so happened that object turnover, competition, and finally, globalisation, enhanced the need of suitable means and rules aimed to extend, or at least to control, the service life of the myriad of objects, facilities, and infrastructure that became available.

Hence, the development of science and technology related to material corrosion and protection (first of all metals) was almost totally ignored in previous centuries due to the elitarian and careful use of the then scarcely available (and expansive) industrial or better artisan products. Corrosion science had a slow and difficult path mainly due to the widely shared misconception of the infinitive availability of both resources and energy. It was only after the second world war that the need for a more "Scrooge" turnover was first evident, and Professor H.H. Uhlig, in 1949, was allowed to demand the UN a strong effort to enhance both knowledge and awareness of the problem.

It is now true and evident that the first aim was reached since most of the mechanisms of corrosion are clear, and the most sensitive and reliable test procedures are established. Unfortunately, the second one is far from being achieved. The cost of corrosion, estimation after estimation, always remains inside the 3%–4% of the GNP of an industrial nation [1]. However, it is estimated that the indirect cost to the end user can double the economic impact. This means that the overall cost to society could be as much as 6 percent of the GDP. Often, the indirect costs are ignored because only the direct costs are paid by the owner/operator.

Of the corrosion-control methods, paints and corrosion-control coatings make up the largest portion [2–4]. Other commonly used methods include the use of corrosion-resistant metals and alloys [5], the application of cathodic and anodic protection [6,7], the use of corrosion inhibitors [8,9], and the use of polymers or composite materials. The cost of corrosion-related services was estimated to be small.

The preventive strategies include:

(1) Increase awareness of high corrosion costs and potential savings;
(2) Change the misconception that nothing can be done about corrosion;
(3) Change policies, regulations, standards, and management practices to increase corrosion cost-savings through sound corrosion management;
(4) Advance design practices for better corrosion management;
(5) Advance life prediction and performance assessment methods;
(6) Advance corrosion technology through research, development, and updating;
(7) Improve education and training of staff in recognition of corrosion control.

A deep gap since ever exists between corrosion science and corrosion control. Another consequent gap is between corrosion control technology and the widespread awareness of its practice. It follows that the priority needs of our field of interest are, mainly, education, and continuing education at all levels of people involved; this is the most important item since it is necessary to spread out knowledge of the very simple corrosion principles to avoid the most elementary mistakes!

During the long and busy career of the authors, the role to be played was to be involved either in a project or in failure analysis. The latter is, unfortunately, the more frequent case. On the alternative, the involvement happened halfway, sometimes on time to correct the design or materials selection mistakes. The avoidable consequences of the lack of basic corrosion control in materials selection and design have been shown in previous papers [10,11]. The subject of the present paper is the case of an avoided disaster that might have happened in a dangerous construction due to the wrong choice of a component, called snap rings or circlips, weighing ten grams. Moreover, the paper will analyze several other construction details where the choice of material, connection, as well as surface finishing, allowed for possible misfunction. These ones did not affect correct function but caused the appearance of ugly corrosion spots.

## 2. Corrosion Problems on Al-Bayt Stadium

### 2.1. Material Solutions in the Stadium

Al Bayt Stadium (Figure 1) (Istād al-Bayt, i.e., 'The House Stadium') is a football stadium in Qatar, which was opened in time for matches in the 2022 FIFA World Cup. The stadium has a retractable roof (Figure 2) that can close completely within 20 min, enabling play throughout the year. The roof weighs around 1600 tons—the equivalent of almost 380 medium-sized motor cars. Each roof truss measures 94.4 m and weighs between 82 and 104 tons. It is possible to close the roof at the touch of a button, in a three-phase movement. The movement for locking/unlocking the trolleys is obtained through sixty-four actuators (Figure 3), a couple at the end of each one of the 32 trolleys of the movable ceiling. The following (Figure 4) shows the apparatus for the transmission of motion from the motor to the trolley (left) and for blocking the motor (right). Pins allow for the continuity of the structure, and they are kept in place by means of mechanical stops called "snaprings" or "circlips" (Figure 5). A snap ring is a retaining ring consisting of a metal ring with open ends that can be forced into place into a machined groove in order to allow rotation but to prevent lateral movement. They have the ends formed to aid installation and removal and are not formed from wire (i.e., they do not have a round cross-section). These rings are designed to be installed and removed with special pliers.

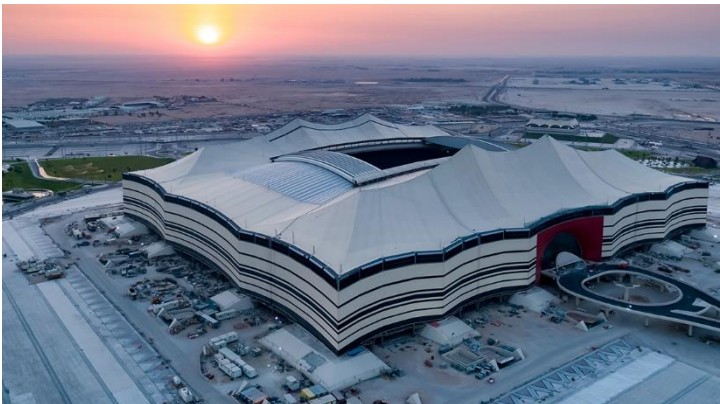

**Figure 1.** The design of the stadium reflects the inside of a Bedouin tent: colored red, white, and black. These tents, the Bayt al sha'ar, are also where the stadium gets its name from. The roof is retractable, opening and closing in 20 min. Al Bayt Stadium has a capacity of roughly 60,000 seats, divided over three tiers.

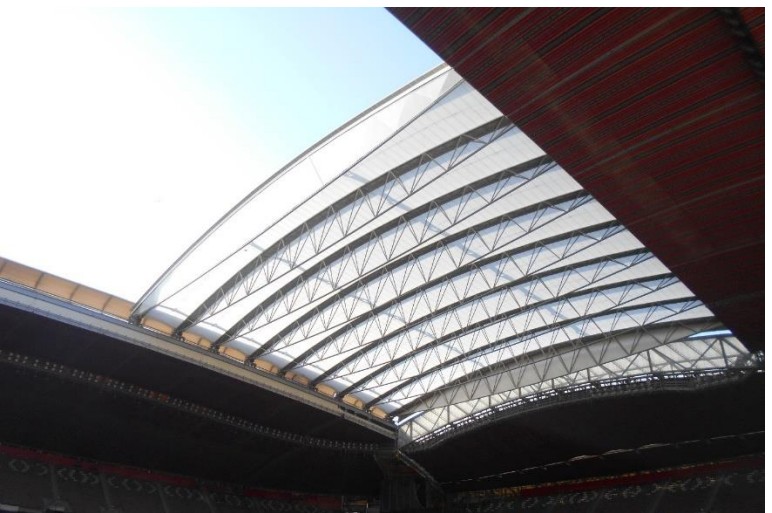

**Figure 2.** The retractable roof built-in polytetrafluoroethylene (PTFE) woven fiberglass membrane is moved by 64 synchronized engines through actuators.

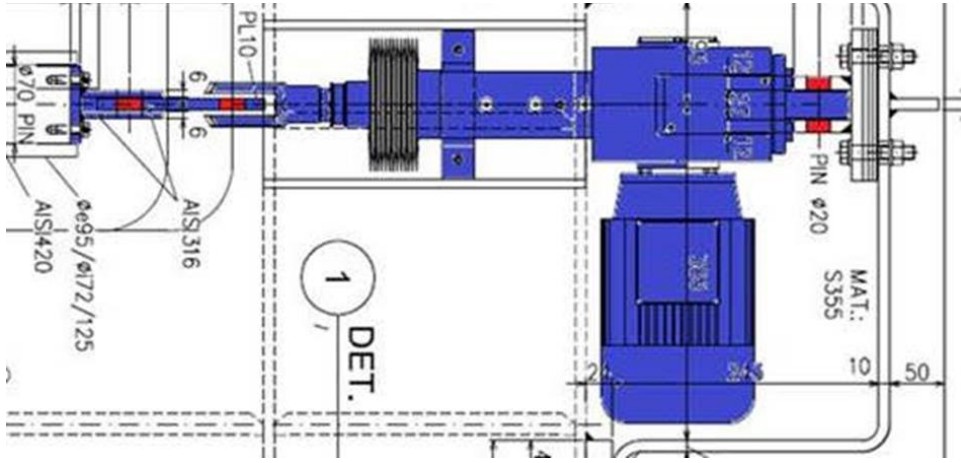

**Figure 3.** Actuator for the rolling/enrolling of the retractable roof with details of the motion transmission devices.

| Layout 1 (left) | Layout 2 (right) |
|---|---|
| 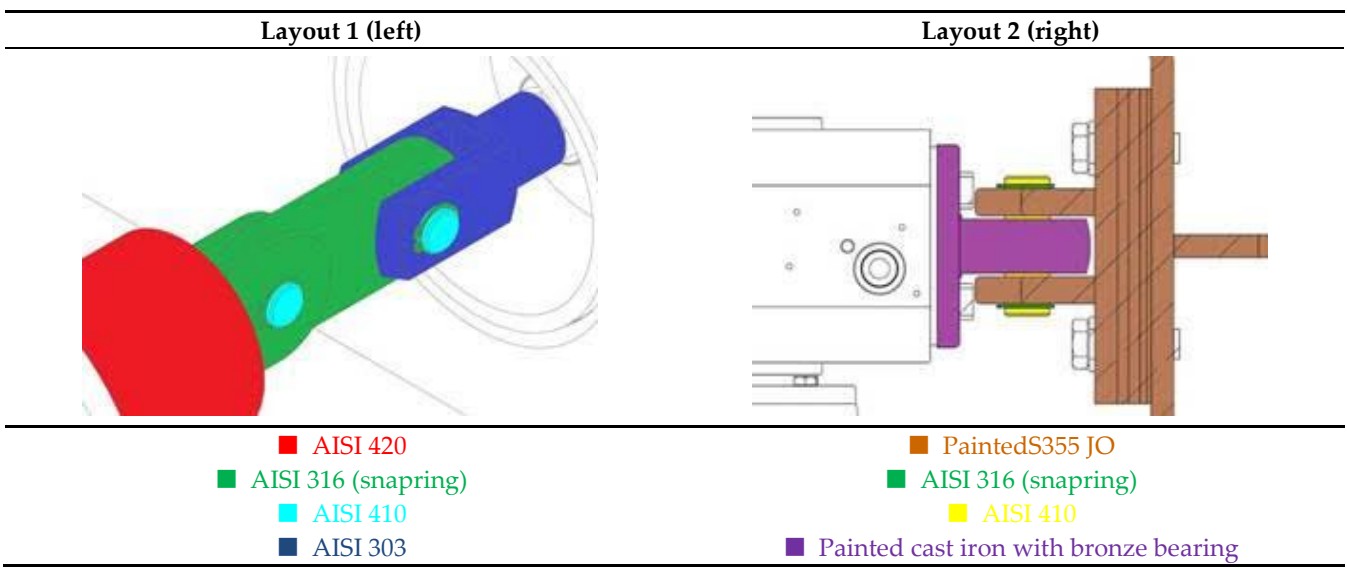 | |
| ■ AISI 420 | ■ PaintedS355 JO |
| ■ AISI 316 (snapring) | ■ AISI 316 (snapring) |
| ■ AISI 410 | ■ AISI 410 |
| ■ AISI 303 | ■ Painted cast iron with bronze bearing |

**Figure 4.** Details of the actuator showing the locations of the snap rings.

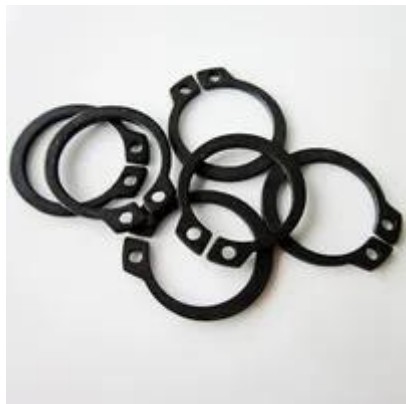

**Figure 5.** Snap rings or circlips.

By design, the 384 pins (six in each actuator) were made from AISI 316 stainless steel. It was observed that many of them showed signs of corrosion and break after a relatively short service period (less than 1 year) (Figure 6).

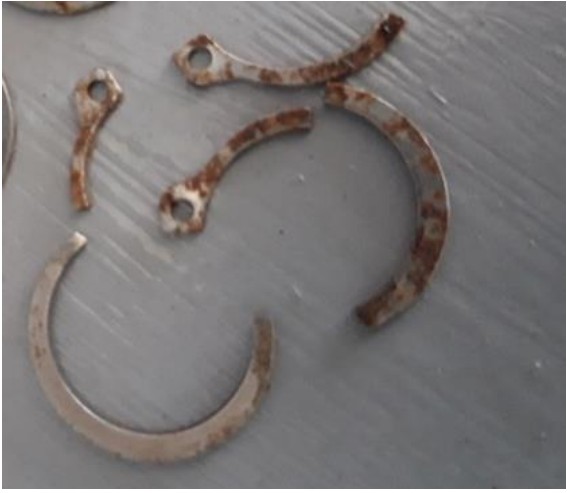

**Figure 6.** Examples of stress corrosion cracked snap rings.

### 2.2. Description of the Corrosion Phenomena in the Stadium

The aim of the present paper is to show how important and sometimes critical is the conscious choice of both materials and their connections, even in the smallest details, especially when such details are present in very large numbers. We took as a significant example the movement mechanisms of the movable roof, in particular, the movements of the actuators for locking/unlocking the trolleys (Figure 7).

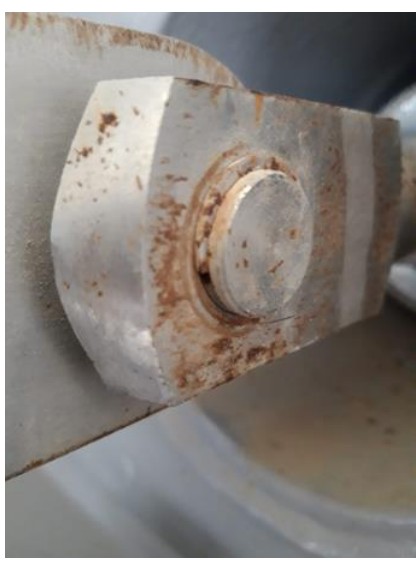

**Figure 7.** Detail of the support roller showing the snap ring located in a carved narrow groove. Localized corrosion and rusting are shown.

### 3. Discussion of the Design Mistakes and of the Causes of Damages

From the analysis of the photos from the study of the existing layouts of the actuator, it appears that the object is composed of a chain of different stainless steels of different compositions and structures with obvious difference in their behavior in terms of corrosion, namely two austenitic (AISI 316 and AISI 303) and two martensitic (AISI 410 and AISI 420) alloys. There is no apparent reason for such a choice, while it is possible for a designer to consult many sources of information (databases) about the suitable metallic material for a given use [12–15].

Martensitic stainless steels have a reduced and competitive price and high mechanical strength when compared to other classes of stainless steel. The yield strength ranges from 275 MPa in annealed condition up to 1900 MPa in the quenched and tempered condition. Comparing AISI 410 with AISI 420, we learn that both are hardenable, with 420 reaching the highest hardness combined with good toughness. Both show corrosion resistance in the atmosphere, fresh water, steam, oil, petrol, alcohol, and organic solvents, they resist heat oxidation up to 650 °C (furnace parts, turbine blades, burners, and valve parts) and they exhibit good weldability. They are used in the construction of components for hydroelectric, petrochemical, and general mechanical plants, such as scissors, knives, measuring instruments, flanges and fittings, etc.

This couple of austenitic stainless steels, AISI316 and AISI303, do not allow a selective comparison between each other since the differences in both features and performance is very high. The 316 achieves increased mechanical strength through cold deformation; it has good weldability, good corrosion resistance to a wide variety of salts and organic acids, and fair-to-weak solutions of reducing acids. It resists intergranular corrosion up to 300 °C. It is used in the petroleum, chemical, food, textile industries and in parts for ovens and naval equipment.

AISI 303 includes a high sulfur content to improve its mechanical processing. Corrosion resistance is reduced due to the presence of sulfur but still satisfactory in the

atmosphere and towards foodstuffs and organic chemical products. It is not solderable. It is used for working on high-speed machines in the production of pins, bushings, screws, nuts, tie rods, etc. It follows that the use of both, in terms of performance and cost, is not justified in comparison with the suitable and cheaper AISI 410 or 420 [16].

Figure 7 shows that the object is affected by the presence of generalized rusting and widespread pitting phenomena, denoting a particularly aggressive environment, the effects of which are enhanced by some mistakes in both design and construction, namely:

(a) The object is composed of a chain of different stainless steels of different compositions and structures with obvious differences in behavior in terms of corrosion [5,12]: AISI 316 steel is the noblest one and acts as a cathode compared to AISI 303 steel, which is not recommended for use in highly corrosive environments due to the presence of sulfur in the composition.

(b) The electrochemical corrosion due to the presence of a galvanic chain of different corrosion potentials of the differently "noble" components might have been minimized by the peculiar property of stainless steels to be "passivable alloys": the main feature of all stainless steels is that they are suitable for building a nanometric surface layer consisting mainly of chromium oxide, named the passive layer. It is self-healing, and the rate of restoring passivity after damage is a feature of the quality of the alloy. Thus, the stability of the passive film is the result of a compromise between two kinetic processes: film growth and film dissolution [17]. Moreover, the surface stationary electrochemical conditions are driven mainly by adsorption/desorption equilibria of external substances rather than via redox electrochemical reactions, the reason why for passive stainless steels an electrochemical corrosion potential does not exist. In the active state, on the other hand, stainless steels have an often higher corrosion rate than carbon steels. Nevertheless, heat treatments, mechanical working, and handling produce many surface defects, hence reducing the corrosion resistance of the passive layer. To increase and optimize the formation of the chromium oxide layer, it is necessary to perform a repassivation procedure. The immersion of stainless steel in an acid bath dissolves free iron or other foreign substances from the surface while leaving the chromium intact, according to ASTM380 [18]. The acid chemically removes the free iron, leaving behind a uniform surface with a higher proportion of chromium than the underlying material [19]. This treatment was not performed on the structure. In our case, the AISI 303 s.s, both for the galvanic coupling and for the poor finish, is subject to widespread pitting.

(c) Poor surface finishing is the third construction mistake. Both reliability and durability of the passive layer are strictly connected with the smoothness of the surface; it is not obviously possible to reach a perfectly planar surface, and at the crystallographic level, all surfaces are rough. Hence, passivity is a dynamic situation where activation phenomena happen in a very high number of sites, quickly followed by repassivation. The speed of reestablishing passivity is a function of both surface uniformity and quality. All functional accessories of the actuator (the colored parts of Figure 4) have been produced in the mechanical workshop leaving the degree of finish produced by the tool. However, this roughness deviation from the optimum could have been minimized by a suitable repassivation process.

(d) All edges should be rounded, not only to allow more uniform distribution of possible coatings but also to avoid the accumulation of current density on sharp edges, enhancing both localized rust and pitting, as clearly shown in Figure 7.

(e) Moreover, the designer coated the cast iron and SS355JO steel components, which are anodic, to both the joined S.S. and the bronze bearings: huge cathode, small anode, high corrosion current density through the pores of the paint (see the archetypic example of such a mistake in the book: Corrosion Engineering, by M. Fontana, N. Green) [20].

Regardless of the above-mentioned phenomena, the most pressing problem concerns the behavior of the snap rings located in a narrow groove carved on the support roller

for the movement of the mechanism (Figure 5), which is built from AISI 410 steel with good resistance characteristics, even if it is not recommended to use in highly corrosive environments. It is, however, anodic with respect to the material used for the snap rings (AISI 316), and, in principle, it might provide the snap ring with some form of cathodic protection. The galvanic chain is, however, irrelevant in this case, in view of another phenomenon much more serious and directly responsible for the early breaking of the snap rings. It is dependent on construction and the aggressiveness of the environment and is called crevice corrosion. Crevice corrosion [21,22] occurs in those hidden sites where oxygen does not circulate freely (Figure 8).

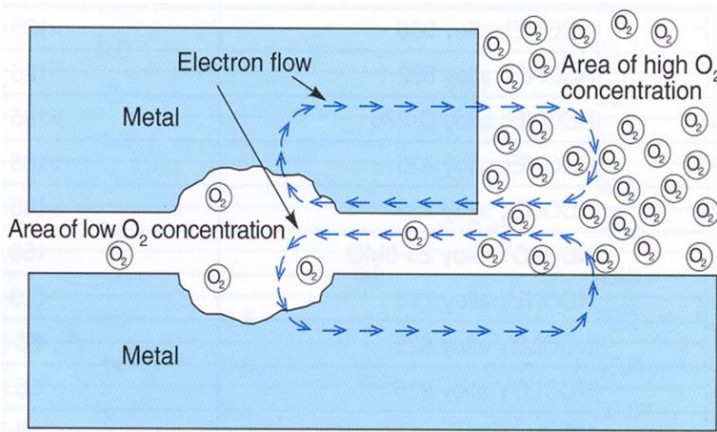

**Figure 8.** Mechanism of crevice corrosion.

In these areas, oxygen is progressively consumed and cannot be renewed by diffusion or convection. Outside the confined area, the cathodic reaction of oxygen reduction still occurs on the metal surface, whereas, in the confined area, the metal surface becomes the anode. The corrosion processes lead to the modification of the chemical composition of the electrolyte in the crevice with simultaneous acidification and an increase of chloride ion concentration. When the pH and the Cl− concentration in the crevice solution reach critical values, depassivation of the stainless steel with active dissolution occurs, which is localized to the points where passivation is weakest, i.e., in the anodic areas. In this specific case, the snap ring was designed to be built using AISI 316, a passive alloy that critically depends on the passive ceramic oxide layer that covers it and which, by default, should reform itself in a very short time in case of damage. The ring is inserted under tension in the thin groove engraved on the cylinder. Inside the cavity, the conditions described above are very suitable for crevice corrosion. The metal attack occurs in those more anodic areas, that is, where the oxide is most damaged and the metal structure most weakened. These areas are located where the effects of the applied mechanical tension are more pronounced, and localized self-stimulating corrosion will begin until (in a very short time) the crack formed does not produce the fracture of the arc according to the mechanism of corrosion under tension (stress corrosion). This mechanism has been thoroughly explained by J.C Scully in a classic publication [23], showing that the events occurring at the tip of a propagating stress corrosion crack depend on the relative prevailing contribution of the strain rate and repassivation rate (Figure 9). It is argued that the strain rate and repassivation rate interact to maintain an acidified solution at the tip of a propagating crack in halide solutions just below the pitting potential. If the strain rate is too low or the repassivation rate too large, repassivation will occur, resulting in crack arrest. On the contrary, the crack will form and proceed at a rate that is self-accelerated due to the corrosion cell produced by the anodic tip and the other cathodic repassivating surfaces.

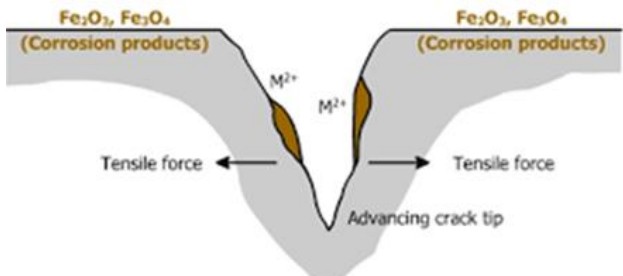

**Figure 9.** Mechanism of stress corrosion.

Therefore, there is an overlap between the most dangerous corrosion conditions: a cell with differential aeration: oxygen-rich external cathodic areas, asphyctic, acidic, chloride-rich anodic internal areas; breakdown of local passivity producing pitting; and applied mechanical tension giving rise to stress corrosion. Under such conditions, it could be deduced that a non-passivated spring steel snap ring could resist longer because the attack would be distributed over the entire surface of the snap ring with lower current density and much less risk of stress corrosion. The unknown factor is given by the corrosion rate (rust and general thinning) of the metal. It is, therefore, better to choose an alternative alloy with a low corrosion rate.

## 4. Solutions to Reduce Further Corrosion

Considering the critical role of the snap rings, the safest possible solution is the use of beryllium bronze, a non-passivated material with a much greater resistance to generalized corrosion than spring steel. However, the problem of the cavity remains, which causes the differential aeration cell to act. Stressed bronze, although not passivated, nevertheless presents areas with less noble potential where the tension accumulates, and the risk of preferentially localized breakage remains.

It is then possible to try to inert all the elements of the cell by means of surface protection. By covering both the slot and the snap ring, both the anode and cathode can be disengaged. The problem now remains limited to the duration of protection provided by the coating. The operating conditions exclude any painting, with the risk of staining the white tissue of the ceiling.

The only possibility is the use of viscous protecting creams widely used in automotive building procedures, for example, in spot-welded sheets or in the hidden underbody. Those creams are very adherent, do not drip, and contain products such as Teflon and lithium soaps, which ensure its stability over time, therefore allowing for five-year monitoring.

The procedure should be as follows:

- Disassembly of the snap ring;
- Cleaning the seat and adjacent areas with a bronze brush;
- Spraying of the entire surface area of the roller;
- Spraying or immersion of the snap ring;
- Assembly of the snap ring.

The high viscosity of the product should avoid any dripping or pollution of the surrounding areas, even for temperatures exceeding 70 °C.

In principle, this solution could avoid the use of more noble materials than spring steel. However, as they used to say: belt and suspenders... Therefore, the final solution was beryllium copper.

## 5. Conclusive Remark

It seemed to the authors that the analysis of the components at the level of functional detail described above represents the paradigm of the infinite combinations of materials, applied stresses, protection methods, the forecasting of durability, and the innovative solutions that the designer encounters in large infrastructural works. Before assuming any

final decision, it should be necessary to consult an expert in anticorrosion since general and basic knowledge relating to this matter is often a source of innocent mistakes due to the right choice being anti-instinctive. In our case, e.g., the concept of "stainless steel" often neglects the existence of thousands of passivable alloys with a wide performance range.

Luckily, all of the above-mentioned negligence were venial sins without significant effects on durability but undoubtedly affecting aesthetics. Luckily, again, the fracture of the AISI 316 snap rings was so quick that it happened while the stadium was still under construction, and a remedy was found without harm.

**Author Contributions:** Conceptualization, P.L.B.; Investigation, A.K.; Data curation, A.K. and P.L.B.; Writing—original draft, A.K.; Writing—review & editing, P.L.B. All authors have read and agreed to the published version of the manuscript.

**Funding:** This research received no external funding.

**Institutional Review Board Statement:** Not applicable.

**Informed Consent Statement:** Not applicable.

**Data Availability Statement:** The data is available on reasonable request from the corresponding author.

**Conflicts of Interest:** The authors declare no conflict of interest.

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
