# Peer review of "Corrosion Hazards in Urban Infrastructure Structures Using the Example of the Al Bayt Stadium in Katar"

_coatings, doi:10.3390/coatings13081443_

Round 1
Reviewer 1 Report
The manuscript deals with explanation of one corrosion example from the practice but the analysis is mainly performed based on qualitative assessment of materials without some exact data, such as corrosion potentials or similar. It would be useful to provide some quantitative data to improve the discussion.
Authors describe seegers as retaining rings but what can be found in literature that Seeger is a company producing retaining rings. Please check the terminology.
Elements presented in Figure 4 named as seeger do not look like rings so it is hard to imagine where are the rings from figure 6 actually placed.
Line 13 - It is not clear what does it mean "consequences of such mistakes were so precocious" please rewrite
Figure 1 – please delete the last sentence in figure description as it is not relevant for the subject of this work
Line 179- how did authors check the surface finishing (roughness)?
The manuscript need some improvement regarding the language with some remaining typing errors and ambigous sentences, especially in the abstract. I would suggest avoiding terms such as ugly, happily..
Reviewer 2 Report
There are the following comments:
1. Lines 173,174. It should be explain the expression "free iron".
2. Fig. 9. It is not clear which mechanism of SCC illustrates this drawing. The “classic” mechanism of SCC of stainless steels is a rupture of a passive film in the top of the crack with the subsequent dissolution of the metal.
